# Acceleration from a clustering environment

Roi Holtzman[1] and Christian Maes[2]

[1]*Physics of Complex Systems, Weizmann Institute of Science, Rehovot, Israel*

[2]*Department of Physics and Astronomy, KU Leuven, Belgium*

We study the effects of correlations in a random environment on a random walker. The dependence of its asymptotic speed on the correlations is a nonperturbative effect as it is not captured by a homogeneous version of the same environment. For a slowly cooling environment, the buildup of correlations modifies the walker's speed and, by so, realizes acceleration. We remark on the possible relevance in the discussion of cosmic acceleration as traditionally started from the Friedmann equations, which, from a statistical mechanical point of view, would amount to a mean-field approximation. Our environment is much simpler though, with transition rates sampled from the one-dimensional Ising model and allowing exact results and detailed velocity characteristics.

## I. INTRODUCTION

The motion of a colloidal particle suspended in a fluid may be well-described by summarizing its environment in terms of its temperature and viscosity and a few other macroscopic parameters, such as its density profile. However, stronger coupling and nontrivial correlations in the environment can bring complications, and careful consideration is required in determining the features to be taken into account. For example, a probe embedded in an equilibrium bath or in an active bath behaves differently, even when both baths have the same density [1]. Similarly, the motion of a random walker embedded in a biased random potential is affected by the correlations in the potential. Averaging the correlations out and keeping only the bias, as in mean-field approaches, may lead to an improper description of the walker.

In the present paper, we study the above warning in great detail for exactly solvable models. We take Ising spins as an environment for a biased one-dimensional random walker. Its asymptotic or stationary speed is self-averaging and we compute that speed as a function

of temperature for a fixed magnetization. We discover a rich variety of possible behaviors, including thresholds and non-monotone behavior of the speed in a cooling environment. Therefore, slowly changing the Ising temperature, slower than it takes the walker to achieve steady state, which amounts to modifying the correlation length, induces a change in speed or an acceleration in the probe's movement. That would not be visible in case the environment is described solely by the magnetization which amounts to an annealed version of the setup.

That brings us to a particular motivation of the paper, which concerns the ongoing debate (or controversy) regarding the origin or the nature of *dark energy*. One of the main reasons to believe there is truly something like 'dark energy' (and not only in an effective way) lines up, so it appears to the authors, with an over-reliance on the Friedmann equations of general relativity. We speculate that those Friedman equations represent an annealed average, which washes out the disorder and yields a non-accelerating expansion of the Universe. Yet, when looking at a typical disorder realization of the Universe, we may find an accelerating expansion even without the explicit introduction of a cosmological constant. That scenario is necessarily nonperturbative.

The literature on random walks in a random environment is vast. Both from the probability and from the statistical mechanics side, a wide plethora of models have been studied, and much has been understood and solved, especially in one dimension. We cannot possibly include all original sources, but we mention the reviews by Zeitouni for the mathematical theory, [2, 3]. A large literature is also devoted to the diffusion of random walks in a random environment. If the average work done on the walker is zero, there can be ultra-slow diffusion, such as in the Sinai model, [4]. For our purpose, as we are mainly interested in the form of the asymptotic velocity, we follow the work of Derrida [5]. Somewhat related is also the extensive numerical study for a Brownian particle in a random potential, [6].

The present paper is concerned with the velocity of the walker in a random force field, where the force involves Ising-spins and is both thermally and athermally biased. The forcing at different sites is correlated, and that correlation structure gives a rich behavior of how the walker is affected by slow cooling of the environment. The reported results are nonperturbative and exact.

Plan of the paper: We start in the next Section II with a summary of Derrida's result for the stationary velocity of a random walker in a random environment. The analysis is done for a random walk in the Ising model environment. The Ising configuration $\sigma$ is sampled with the standard nearest neighbors coupling and in a magnetic field. There are two cases, Section III when the work on the walker to jump $n \to n + 1$ depends on $\sigma_n$, and Section IV where the work depends on $\sigma_n \sigma_{n+1}$. Each time, we present a full analysis of the influence of correlations and parameters on the behavior of the velocity as a function of the inverse temperature. The remarks in Section V end with the suggestion that the results of the paper, by analogy, fit in the ambition of the backreaction program for understanding the accelerated expansion of the universe.

## II.   ONE-DIMENSIONAL SETUP

We consider a random walker $X_t$ on the integer sites in continuous time $t$. Hops are allowed only between neighboring sites. The Master Equation describes the evolution of the probability $\text{Prob}[X_t = n] = p_n(t)$ and is given by

$$\frac{dp_n(t)}{dt} = k(n-1, n)p_{n-1}(t) + k(n+1, n)p_{n+1}(t) - (k(n, n-1) + k(n, n+1))\, p_n(t),$$

where the hopping rates $k(n, n \pm 1)$ from site $n$ to site $n \pm 1$ are chosen from a predefined distribution (to be specified later) to which we refer as the *environment*. We thus have a random walker in a random environment. Averages over this random choice are denoted by angular brackets $\langle \cdot \rangle$.

The work by Derrida [5] provides the steady state velocity $V_N$ of the walker for a finite periodic lattice interval of length $N$,

$$V_N = \frac{N}{\sum_{n=1}^{N} r_n} \left[ 1 - \prod_{n=1}^{N} \left( \frac{k(n+1, n)}{k(n, n+1)} \right) \right] \tag{II.1}$$

$$r_n = \frac{1}{k(n, n+1)} \left[ 1 + \sum_{i=1}^{N-1} \prod_{j=1}^{i} \left( \frac{k(n+j, n+j-1)}{k(n+j, n+j+1)} \right) \right]. \tag{II.2}$$

In many cases of *physically relevant* environments, the velocity $V_N$ is self-averaging in the limit $N \to \infty$, *i.e.*, the limit $t \to \infty$ yielding steady behavior and the limit $N \to \infty$ commute, [7].

Assuming that

$$\left\langle \log \frac{k(n, n+1)}{k(n+1, n)} \right\rangle > 0, \tag{II.3}$$

corresponds to pushing the walker to move to the right. Moreover, condition (II.3) implies

$$\lim_{N \to \infty} \left[ 1 - \prod_{n=1}^{N} \left( \frac{k(n+1, n)}{k(n, n+1)} \right) \right] = 1,$$

and the asymptotic velocity (II.1) becomes

$$V^+ = \lim_{N \to \infty} \frac{1}{\frac{1}{N} \sum_{n=1}^{N} r_n}, \tag{II.4}$$

where we added the superscript "+" to emphasize that the motion of the walker is to the right.

Alternatively, with regard to (II.3), assuming $\left\langle \log \frac{k(n+1, n)}{k(n, n+1)} \right\rangle > 0$ pushes the walker to the left. We then put

$$\ell_n = \frac{1}{k(n+1, n)} \left[ 1 + \sum_{i=1}^{N-1} \prod_{j=1}^{i} \left( \frac{k(n+j, n+j+1)}{k(n+j, n+j-1)} \right) \right] \tag{II.5}$$

$$V^- = - \lim_{N \to \infty} \frac{1}{\frac{1}{N} \sum_{n=1}^{N} \ell_n}, \tag{II.6}$$

where the superscript "−" reminds us that the walker moves to the left.

As an example, we take Eq. 88 of [5], where with probability $\alpha$, $k(n, n+1) = W, k(n+1, n) = 1$, and with probability $1 - \alpha$, $k(n, n+1) = 1, k(n+1, n) = W$ for some $W > 1$. There is a symmetry $\alpha \leftrightarrow 1 - \alpha$. Then, the velocity is nondecreasing as a function of $\alpha$, and the speed is zero for $\alpha \in [1 - \alpha_1, \alpha_1]$, where $\alpha_1 = W/(1 + W)$. The condition $\langle k(n+1, n)/k(n, n+1) \rangle = \alpha/W + W(1 - \alpha) < 1$ is necessary for $V^+ > 0$ and determines $\alpha_1$, while the weaker (II.3) only defines the direction of pushing, not whether the speed is nonzero. Note however that the speed can be zero while the walker is still moving to the right, just not with a displacement proportional to the elapsed time.

Our goal is to map the effect of *correlations* in the environment on the velocity, *i.e.*, on (II.4) or (II.6). For that purpose, we consider the Ising model to represent the environment in the next two sections. This model yields analytic expressions for the velocity, and it allows to control the correlations by varying the temperature.

## III.   ISING SITE DISORDER

In this and the next sections, we use the Ising model to create a disordered environment for the walker but not in the sense that Ising energy differences decide the transition rates; the walker is driven by random force in terms of either single site (this section) or bond (next section) Ising spins.

### A.   Definition of the model

For random (or, disordered) environment, we choose the standard one-dimensional Ising model with Hamiltonian

$$H_N = -J \sum_{i=1}^{N} \sigma_i \sigma_{i+1} - h \sum_{i=1}^{N} \sigma_i, \quad \sigma_i = \pm 1, \tag{III.1}$$

for coupling $J \geq 0$, magnetic field $h$, and periodic boundary conditions, $\sigma_{N+1} = \sigma_1$ with (soon) $N \uparrow \infty$.

The Ising configuration $\sigma$ determines the hopping rates of our walker via

$$k(n, n+1) = e^{+(\beta\varepsilon - a)\sigma_n}, \quad \text{for } n \to n+1, \tag{III.2}$$

$$k(n+1, n) = e^{-(\beta\varepsilon - a)\sigma_n}, \quad \text{for } n+1 \to n, \tag{III.3}$$

where $\varepsilon, a$ are extra parameters that allow tuning the random bias in the behavior of the walker. Note that the spins over the edge $(n, n+1)$ are not treated symmetrically; that will change in Section IV.

We can think of $-\sigma_n$ as a local slope the walker has to overcome in order to hop from $n$ to $n+1$. Interestingly, the bias from $\varepsilon$ and the bias from $a$ may compete. There is the thermal bias $\beta\varepsilon$ where work $\epsilon$ is done and dissipated in a heat bath at inverse temperature $\beta$, while the (dimensionless) parameter $a$ pushes the walker uphill when $a > 0$. That athermal bias with $a$ can be thought to originate from nondissipative effects (such as initial conditions or local forces). Obviously, when $\beta\varepsilon = a$, there is no net motion. In other cases, the velocity can be positive, negative, or zero.

In addition, the environment has a bias as well, determined by the magnetization. To break the symmetry in the environment, we set the magnetic field $h > 0$, and $h$ may depend on $\beta$ even to the extent that $h(\beta)\beta > 0$ as $\beta \downarrow 0$.

Note that the 'local' current is not just given by

$$k(n, n+1) - k(n+1, n) = 2\sigma_n \sinh(\beta\varepsilon - a).$$

If we average the disorder at the level of the transition rates, we get

$$\langle k(n, n \pm 1)\rangle = \cosh(\beta\varepsilon - a) \pm m \, \sinh(\beta\varepsilon - a). \tag{III.4}$$

Using these annealed, averaged (or, homogenized) rates, the current is $2m \, \sinh(\beta\varepsilon - a)$ and obviously only depends on the magnetization $m = \langle\sigma_n\rangle = m(\beta J, \beta h)$ in the Ising model. When we do not average the disorder (but allow self-averaging by passing to the limit $N \uparrow \infty$), we prove that the velocity (or current) depends on the correlations and that it shows an intriguing dependence on the Ising temperature. In particular, that means that the motion may be accelerating for a fixed magnetization when the temperature is changing. We come back to this point in Remark 6 of Section V.

### B. Asymptotic speed

We compute (II.4), and we use the self-averaging property, [5], to have, with probability one,

$$\left(V^+\right)^{-1} = \lim_{N\to\infty} \frac{1}{N} \sum_{n=1}^{N} \langle r_n\rangle_N, \tag{III.5}$$

with the Ising-average $\langle\cdot\rangle_N$ using periodic boundary conditions, $N+1 = 1$ indicated by the subscript $N$.

It is useful to rewrite (II.2) in the form

$$r_n = \sum_{\ell=0}^{N-1} C_\ell(n), \tag{III.6}$$

where $C_0(n) = \frac{1}{k(n,n+1)}$ and for $0 < \ell < N$,

$$C_\ell(n) = \frac{1}{k(n, n+1)} \prod_{j=1}^{\ell} \frac{k(n+j, n+j-1)}{k(n+j, n+j+1)}$$
$$= e^{-2(\beta\varepsilon+a)\sigma_n} \cdots e^{-2(\beta\varepsilon+a)\sigma_{n+\ell-1}} e^{-(\beta\varepsilon+a)\sigma_{n+\ell}}.$$

We need the Ising-average of (III.6) and we observe that

$$\langle C_\ell(n)\rangle_N = \langle C_\ell(1)\rangle_N =: c_\ell(N). \tag{III.7}$$

Therefore, (III.5) equals

$$\left(V^+\right)^{-1} = \lim_{N\to\infty} \frac{1}{N} \sum_{n=1}^{N} \sum_{\ell=0}^{N-1} \langle C_\ell(n)\rangle_N = \lim_{N\to\infty} \frac{1}{N} \sum_{n=1}^{N} \sum_{\ell=0}^{N-1} c_\ell(N)$$

$$= \lim_{N\to\infty} \sum_{\ell=0}^{N-1} c_\ell(N). \tag{III.8}$$

To perform the above sum, we need to calculate the $\ell$-th term

$$c_\ell(N) = \left\langle e^{-2(\beta\varepsilon-a)\sigma_1}\cdots e^{-2(\beta\varepsilon-a)\sigma_\ell} e^{-(\beta\varepsilon-a)\sigma_{\ell+1}}\right\rangle_N = \frac{1}{Z_N}\mathrm{Tr}\left((S_2 M)^\ell S_1 M^{N-\ell}\right), \tag{III.9}$$

where we have introduced the transfer-matrix for the Ising model (III.1),

$$M := \begin{pmatrix} e^{\beta(J+h)} & e^{-\beta J} \\ e^{-\beta J} & e^{\beta(J-h)} \end{pmatrix}, \quad Z_N = \mathrm{Tr}\left(M^N\right)$$

and matrices

$$S_1 := \begin{pmatrix} e^{-(\beta\varepsilon-a)} & 0 \\ 0 & e^{(\beta\varepsilon-a)} \end{pmatrix}, \quad S_2 := \begin{pmatrix} e^{-2(\beta\varepsilon-a)} & 0 \\ 0 & e^{2(\beta\varepsilon-a)} \end{pmatrix}. \tag{III.10}$$

Upon taking the limit $N\to\infty$ of (III.9), a standard calculation yields

$$c_\ell := \lim_{N\to\infty} c_\ell(N) = \frac{e^{-\beta h+3(\beta\varepsilon-a)}}{(\lambda_1-\lambda_2)(\mu_1-\mu_2)}$$

$$\times \left[ \left(\frac{\mu_1}{\lambda_1}\right)^\ell \left(e^{-2(\beta\varepsilon-a)}\mu_1 - \lambda_2\right)\left[e^{\beta J}\left(1-e^{-2(\beta\varepsilon-a)}\right) + e^{\beta h-2(\beta\varepsilon-a)}(\lambda_1-\mu_2)\right]\right.$$

$$\left. - \left(\frac{\mu_2}{\lambda_1}\right)^\ell \left(e^{-2(\beta\varepsilon-a)}\mu_2 - \lambda_2\right)\left[e^{\beta J}\left(1-e^{-2(\beta\varepsilon-a)}\right) + e^{\beta h-2(\beta\varepsilon-a)}(\lambda_1-\mu_1)\right]\right], \tag{III.11}$$

where

$$\lambda_{1,2} = e^{\beta J}\cosh(\beta h) \pm e^{-\beta J}\sqrt{1+e^{4\beta J}\sinh^2(\beta h)} \tag{III.12}$$

$$\mu_{1,2} = e^{\beta J}\cosh(\beta h - 2(\beta\varepsilon-a)) \pm e^{-\beta J}\sqrt{1+e^{4\beta J}\sinh^2(\beta h-2(\beta\varepsilon-a))}. \tag{III.13}$$

The infinite sum (III.8) is a geometric sum in powers of $\mu_1/\lambda_1$ and $\mu_2/\lambda_1$ as can be seen by the explicit expression for $c_\ell$ in (III.11). Since $\mu_1 > \mu_2$, the sum converges for $\mu_1 < \lambda_1$. Thus, the velocity $V^+$ in (III.8) is nonzero if and only if $\mu_1 < \lambda_1$, in which case it equals

$$V^+ = \frac{2e^{2\beta J}\sinh(\beta\varepsilon-a)\sinh(\beta h-(\beta\varepsilon-a))\sqrt{e^{4\beta J}\sinh^2(\beta h)+1}}{e^{4\beta J}\sinh(\beta h)\sinh(\beta h-(\beta\varepsilon-a)) + \cosh(\beta\varepsilon-a)}. \tag{III.14}$$

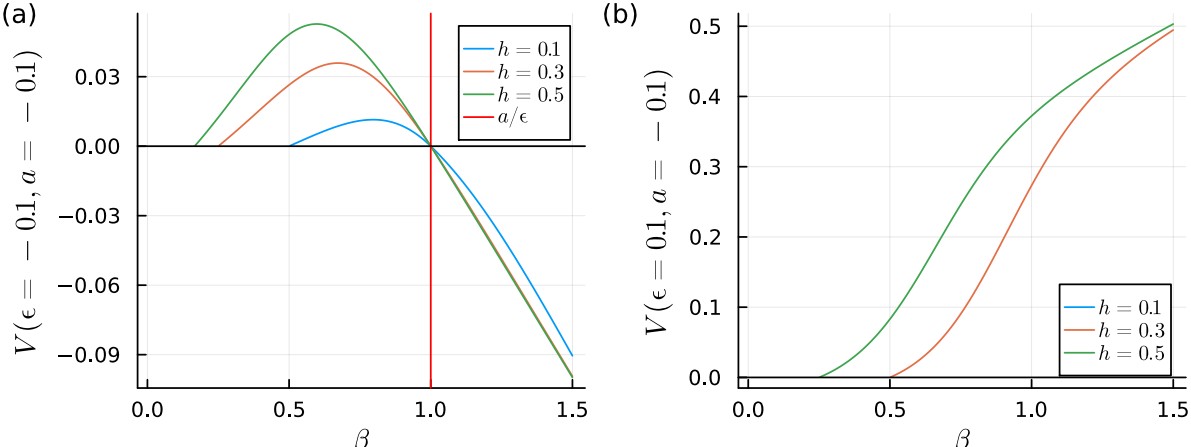

FIG. 1. Velocity $V^\pm$ *vs* inverse temperature $\beta$ for $J = 1$. The left velocity $V^-$ (III.18) is obtained from $V^+$ (III.14) by the mapping $V^-(\varepsilon, a) = -V^+(-\varepsilon, -a)$. Thus, flipping signs of $\varepsilon, a$ simultaneously flips the velocity, and therefore, we consider only cases of competing and noncompeting biases. (a) $\varepsilon = -0.1, a = -0.1$ have the same sign and therefore compete. This results in a non-monotonic behavior of $V^+$. At $\beta = a/\varepsilon$ the velocity vanishes, and for colder temperatures it is negative. (b) $\varepsilon = 0.1, a = -0.1$ are in the same direction, which yields a monotonically increasing velocity with $\beta$. Note that the magnetic field $h$ must be large enough to induce positive $V^+$.

See Fig. 1 for a plot of the velocity as function of $\beta$ for different $a, \varepsilon, h$.

We are interested in the effects of correlations in the environment on the velocity. To decouple the effects of bias and correlations, we want to change $\beta$ while keeping the magnetization $m$ fixed. That requires making the magnetic field $h = h(\beta)$ a function of $\beta$ that keeps $m$ fixed. Since the Ising-magnetization is known,

$$m = \frac{e^{2\beta J} \sinh(\beta h)}{\sqrt{1 + e^{4\beta J} \sinh^2(\beta h)}}, \tag{III.15}$$

the velocity (III.14) can be expressed in terms of that magnetization (valid for $0 < m \le 1$),

$$V^+ = 2\sinh(\beta\epsilon - a) \frac{m - \tanh(\beta\epsilon - a)\sqrt{e^{4\beta J} + m^2(1 - e^{4\beta J})}}{1 - m\tanh(\beta\epsilon - a)\sqrt{e^{4\beta J} + m^2(1 - e^{4\beta J})}}. \tag{III.16}$$

For $m = 0$, the environment is symmetric, and the velocity is zero. At $m = 1$, the velocity becomes $V^+(m = 1) = 2\sinh(\beta\varepsilon - a) = \langle k(n, n+1) \rangle - \langle k(n+1, n) \rangle$: since the system is completely ordered then, averaging at the level of the hopping rates as in (III.4) yields the

correct result.

The correlations appear in the dependence of (III.16) on the coupling $\beta J$, where we recall that the correlation length is

$$\xi \simeq \frac{1}{2}e^{2\beta J}$$

for $\beta J \gg 1$, and $|m| \ll 1$ (low temperature and small magnetic field).

Looking closer at the velocity (III.14), the marginal case is obtained when $\lambda_1 = \mu_1$, with $\lambda_1$ and $\mu_1$ given in (III.12), (III.13). The equality is obtained when $\beta h - 2(\beta\varepsilon - a) = \pm\beta h$. Moreover, $\mu_1 < \lambda_1$ (which yields $V^+ > 0$) only when $0 < \beta\varepsilon - a < \beta h$. Using (III.15), the last inequality can be written as $e^{2\beta J}\sinh(\beta\varepsilon - a) < \frac{m}{\sqrt{1-m^2}}$.

Summarizing: it is necessary and sufficient for $V^+ > 0$ that

$$a < \beta\varepsilon, \quad \text{and} \quad f(\beta) := e^{2\beta J}\sinh(\beta\varepsilon - a) < \frac{m}{\sqrt{1 - m^2}} \tag{III.17}$$

(Note that the first inequality is the one obtained by (II.3): plugging in the hopping rates (III.2) yields $-2(\beta\varepsilon - a)\langle\sigma_n\rangle < 0$, which for $h > 0$ reduces to $\beta\varepsilon - a > 0$.)

The case of $V^- \leq 0$ is analyzed similarly. Following the same procedure as above, we find (recall (III.14))

$$V^- = -\frac{2e^{2\beta J}\sinh(a - \beta\varepsilon)\sinh(\beta h + (\beta\varepsilon - a))\sqrt{e^{4\beta J}\sinh^2(\beta h) + 1}}{e^{4\beta J}\sinh(\beta h)\sinh(\beta h + (\beta\varepsilon - a)) + \cosh(\beta\varepsilon - a)}, \tag{III.18}$$

or, in terms of magnetization (see (III.16))

$$V^- = -2\sinh(a - \beta\epsilon)\frac{m - \tanh(a - \beta\epsilon)\sqrt{e^{4\beta J} + m^2(1 - e^{4\beta J})}}{1 - m\tanh(a - \beta\epsilon)\sqrt{e^{4\beta J} + m^2(1 - e^{4\beta J})}}. \tag{III.19}$$

The conditions for nonzero $V^- < 0$ remain $\mu_1^- < \lambda_1$, where $\mu_1^- = \mu_1(\varepsilon \mapsto -\varepsilon, a \mapsto -a)$ in (III.13). The same arguments that were used in the case of $V^+$ apply here and the necessary and sufficient conditions for $V^- < 0$ are (compare with (III.17))

$$a > \beta\varepsilon, \quad \text{and} \quad f(\beta) > \frac{m}{\sqrt{1 - m^2}}. \tag{III.20}$$

## C.  Discussion

In this Section we analyze the behavior of the velocity from the analytic expression of $V^\pm$ for fixed $m$ and changing $\beta$, (III.16).

Consider first the case $\varepsilon = 0$ and the walker moving to the right. The conditions (III.17) simplify to $a < 0$ and $e^{2\beta J} \sinh(-a) < \frac{m}{\sqrt{1-m^2}}$. Therefore, $V^+ > 0$ if and only if $a < 0$ and $\beta_a := \frac{1}{2J} \log\left[\frac{m}{\sinh(|a|)\sqrt{1-m^2}}\right] > 0$. It implies that there is a nonzero speed $V^+$ only at sufficiently high temperatures ($0 \leq \beta < \beta_a$) and the velocity to the right is zero, even at infinite temperature, when the magnetization $m$ is too small (when $m < \sinh(|a|)/\cosh(a) = \tanh(|a|)$). The velocity $V^+$ (still for $\varepsilon = 0$) is plotted in Fig. 2(a). We see it decreases when lowering the temperature. Indeed, increasing correlations in the environment may create larger domains that oppose the walker's motion to the right, thereby reducing its velocity. For large enough correlations, these domains trap the walker to a halt. Analyzing the motion to the left corresponding to conditions (III.20), yields similarly that $V^- < 0$ if and only if $a > 0$ and $0 \leq \beta < \beta_a$.

For the case $a = 0$ and motion to the right, the conditions (III.17) simplify to $0 < \beta\varepsilon$ and $e^{2\beta J} \sinh(\beta\varepsilon) < \frac{m}{\sqrt{1-m^2}}$. Similar to the previous case, there is now a $\beta_m > 0$ such that $V^+ > 0$ if and only if $0 < \beta < \beta_m$. Note that for $\beta = 0$ the velocity is zero, simply because the hopping rates (III.2) are symmetric for $a = 0, \beta = 0$. As $\beta$ increases, the hopping rates (III.2) become more asymmetric, inducing a nonzero velocity. On the other hand, correlations in the environment increase and reduce the velocity. These competing effects yield a velocity that depends non-monotonically on $\beta$, shown in Fig. 2(b). The case of motion to the left results in the same conditions and the same form of the (negative) velocity.

The behavior becomes even richer when both $a$ and $\varepsilon$ are nonzero. We analyze in detail the case of motion to the right $V^+$ with the conditions (III.17); the case of motion to the left $V^-$ is obtained from the very same considerations but starting with the conditions (III.20).

Consider first the case $a > 0$. Our first condition in (III.17), $\beta\varepsilon > a$, implies that for $V^+ > 0$ we must have $\varepsilon > 0$ and $\beta > a/\varepsilon$ (needing sufficiently low temperatures). The value $a/\varepsilon$ is a positive threshold for $\beta$ to have a strictly positive velocity, and applies for any value of $m$. The second condition in (III.17) yields another threshold temperature, $\beta_m(\varepsilon, a)$, a

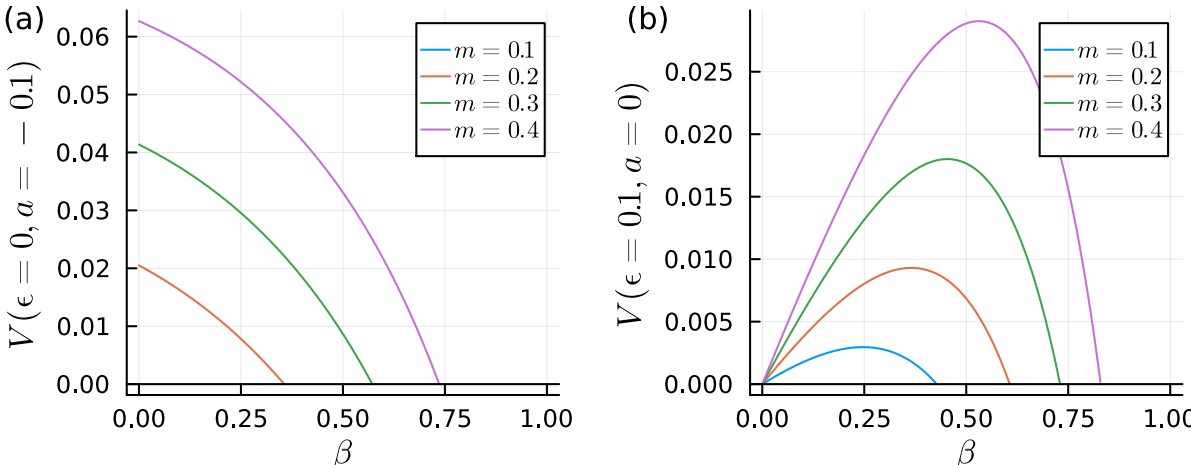

FIG. 2. The velocity as a function of $\beta$, for fixed magnetization, where $J = 1$. (a): $\varepsilon = 0, a = -0.1$ yields a decreasing velocity with $\beta$. The threshold magnetization for $V^+ > 0$ is $m_a = \tanh(|a|) \approx$ 0.1. (b): $\varepsilon = 0.1, a = 0$ produces a non-monotone behavior of the velocity.

function of both $\varepsilon$ and $a$ that solves

$$f(\beta) = e^{2\beta J} \sinh(\beta\varepsilon - a) = \frac{m}{\sqrt{1 - m^2}}. \qquad \text{(III.21)}$$

Clearly, (as we assume $m > 0$), we must have $\beta_m(\varepsilon, a) > a/\varepsilon$.

Summarizing, we have that for $a > 0, \varepsilon < 0$ the velocity is zero, whereas for $a > 0, \varepsilon > 0$, the velocity is positive in some range $\beta \in (a/\varepsilon, \beta_m(\varepsilon, a))$. See inset of Fig. 3(a).

In the case $a < 0, \varepsilon < 0$, the first condition of (III.17) requires $\beta < a/\varepsilon$. That cutoff value of the velocity is independent of $m$ (as indicated by the red line in Fig. 3(b)). For the second condition in (III.17), we note that for the threshold $\beta$, (III.21), there could now be zero, one or two solutions that are smaller than $a/\varepsilon$. The value of $m$ determines the number of solutions. This can be understood by the behavior of $f(\beta)$ in the range $(0, a/\varepsilon)$: It is strictly positive, and at the edges it attains the values $f(0) = -\sinh(a) > 0$ and $f(a/\varepsilon) = 0$. Simple examination of $f(\beta)$ shows it has a single maximum $\beta_{\max}$ in $(0, a/\varepsilon)$, and therefore we find the following three possibilities. If $\alpha_m := m/\sqrt{1 - m^2} < -\sinh(a)$ there is a single solution $\beta_{m_2}$ such that $V^+ > 0$ in the range $(\beta_{m_2}, a/\varepsilon)$. (The threshold determining if there are one or two solutions can be rearranged as $m_a := \tanh(|a|)$, such that for any $m < m_a$ there is a single solution.) If $-\sinh(a) < \alpha_m < f(\beta_{\max})$, there are two solutions, $\beta_{m_1} < \beta_{m_2}$ such that $V^+ > 0$ in two separate ranges $(0, \beta_{m_1})$ and $(\beta_{m_2}, a/\varepsilon)$. Lastly, if $f(\beta_{\max}) < \alpha_m$,

then $V^+ > 0$ for the entire range $(0, a/\varepsilon)$. These cases are all shown in Fig. 3(b) in blue, orange, and green, respectively. Note that the region $[\beta_{m_1}(\varepsilon), \beta_{m_2}(\varepsilon)]$ where $V^+ = 0$ also applies to left motion, namely $V^- = 0$ in this region as well; see Fig. 3(b). For large values of $m$ where there is no solution to (III.21), (such that $\alpha_m > f(\beta_{\max})$) the velocity can also exhibit non-monotonic behavior; see $e.g.$, the green line for $m = 0.35$ in Fig. 3(b).

Lastly, in the case $a < 0, \varepsilon > 0$, the first condition in (III.17) always holds, and the second condition gives a threshold $\beta_m$ that solves (III.21). The solution $\beta_m$ is positive when $|a|$ is small enough: if $\sinh(-a) < m/\sqrt{1 - m^2}$, there is a solution $\beta_m(\varepsilon) > 0$ such that for any $\beta \in [0, \beta_m(\varepsilon))$ the velocity $V^+$ is positive. This threshold magnetization is given by $m = \tanh(|a|)$; see Fig. 4(a). We conclude that the case $a < 0, \varepsilon > 0$ is similar to the case $a = 0, \varepsilon > 0$ plotted in Fig. 2(b), only shifted to the left.

The walker also exhibits motion to the left, see Figs. 3, 4. Adjusting the arguments presented in the paragraphs above for $V^- < 0$, given in conditions (III.20), is straightforward and results in a complementary range of $\beta$ by mapping $\varepsilon \mapsto -\varepsilon, a \mapsto -a$, as can be seen by comparing panels (a), (b) in Figs. 3, 4.

## IV. ISING BOND DISORDER

We consider again the same disordered environment as sampled from the one-dimensional Ising model. However, in contrast with (III.2) where a single local spin value is used, this time, the local coupling energies enter. That is, we take hopping rates

$$k(n, n + 1) = e^{(\beta\epsilon - a)\,\sigma_n\sigma_{n+1}}$$
$$k(n + 1, n) = e^{-(\beta\epsilon - a)\,\sigma_n\sigma_{n+1}}. \tag{IV.1}$$

When the spins $\sigma_n = \sigma_{n=1}$ over the edge agree, the particle prefers to move to the right when $\beta\epsilon > a$ and to the left when $\beta\epsilon < a$. We refer to this model as the "bond model" since the bias is determined by the local interaction energy $i.e.$, the alignment between neighboring spins. Therefore, nonzero velocity can appear even at $h = 0$.

The same procedure as in the previous Section allows calculating explicitly the velocity

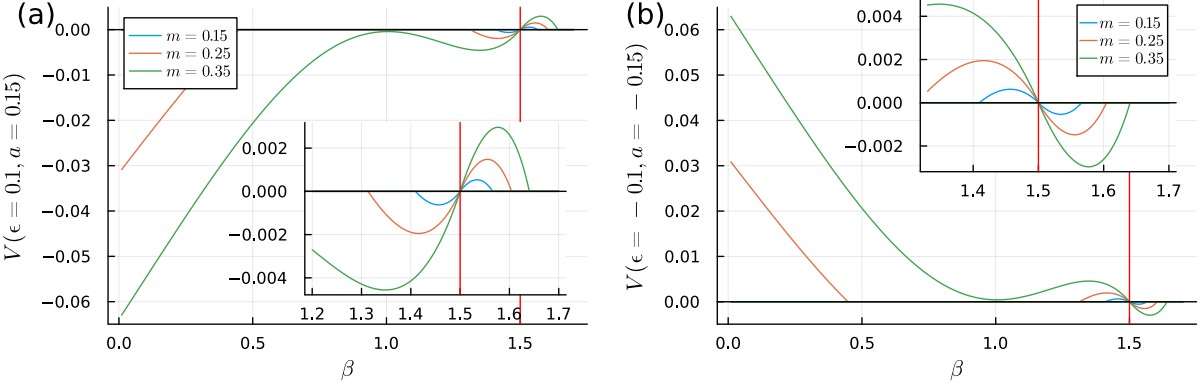

FIG. 3. Velocity $V^{\pm}$ as a function of $\beta$, for fixed magnetization $m$, where $J = 1$. In both plots, the biases $\varepsilon$ and $a$ compete. Insets magnify the region around $\beta = a/\varepsilon$ marked by the vertical red line. (a): $\varepsilon = 0.1, a = 0.15$ yields a non-monotonic velocity $V^+$ with $\beta$, nonzero in $(a/\varepsilon, \beta_m)$. (b): $\varepsilon = -0.1, a = -0.15$, yields nonzero velocity already at $\beta = 0$. For any value of $m$ the velocity vanishes at $\beta = a/\varepsilon$. For large values of $m$ the velocity is decreasing in $\beta$. Yet, for smaller $m$, $V^+$ can be non-monotone, and at low enough values of $m$, the velocity can even vanish at some value $\beta_{m_1}$ (that depends on $m$), and then reappear at $\beta_{m_2}$. The symmetry $a \mapsto -a, \varepsilon \mapsto -\varepsilon$ corresponding to $V^+ \mapsto -V^-, V^- \mapsto -V^+$ is apparent from comparing (a) and (b).

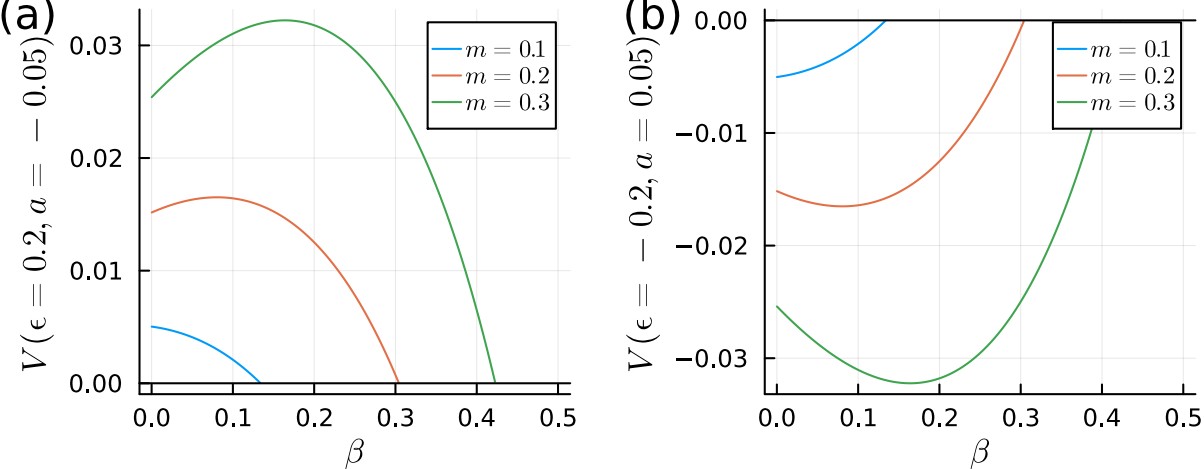

FIG. 4. Velocity $V^+, V^-$ as a function of $\beta$, for fixed magnetization $m$, where $J = 1$. The two plots correspond to noncompeting $\varepsilon$ and $a$. (a): $\varepsilon = 0.2, a = -0.05$ results only in motion to the right, i.e., $V^-$ is zero for any temperature. If $m < m_a := \tanh(|a|) \approx 0.05$, then $V^+ = 0$ for any $\beta$. However, for $m > m_a$, there is a range $[0, \beta_m(\varepsilon))$ where $V^+ > 0$. For large enough values of $m$, $V^+$ is non-monotone. (b): $\varepsilon = -0.2, a = 0.05$, yields a symmetric picture, where $V^+ \mapsto -V^-$ and $V^- \mapsto -V^+$.

as a function of the parameters $\epsilon, a, \beta, J, h$:

$$V^+ = \frac{e^{4b}\left(\sqrt{e^{4K}\sinh^2 H + 1} - 2e^{b+2K}\sinh b \cosh H\right)^2 - e^{8b+4K}\sinh^2 H - 1}{2e^{3b+2K}\cosh H \frac{e^{2b+4K}\sinh^2 H + 1}{\sqrt{e^{4K}\sinh^2 H + 1}} + e^{5b+4K}\left(\cosh(2H) - e^{2b}\right) + 2e^{2b}\cosh b}, \qquad \text{(IV.2)}$$

where $b = -\beta\epsilon + a, K = \beta J, H = \beta h$. The simpler case of only a single bias, namely $\varepsilon = 0$ or $a = 0$, is plotted in Fig. 5, whereas the general case is plotted in Fig. 6.

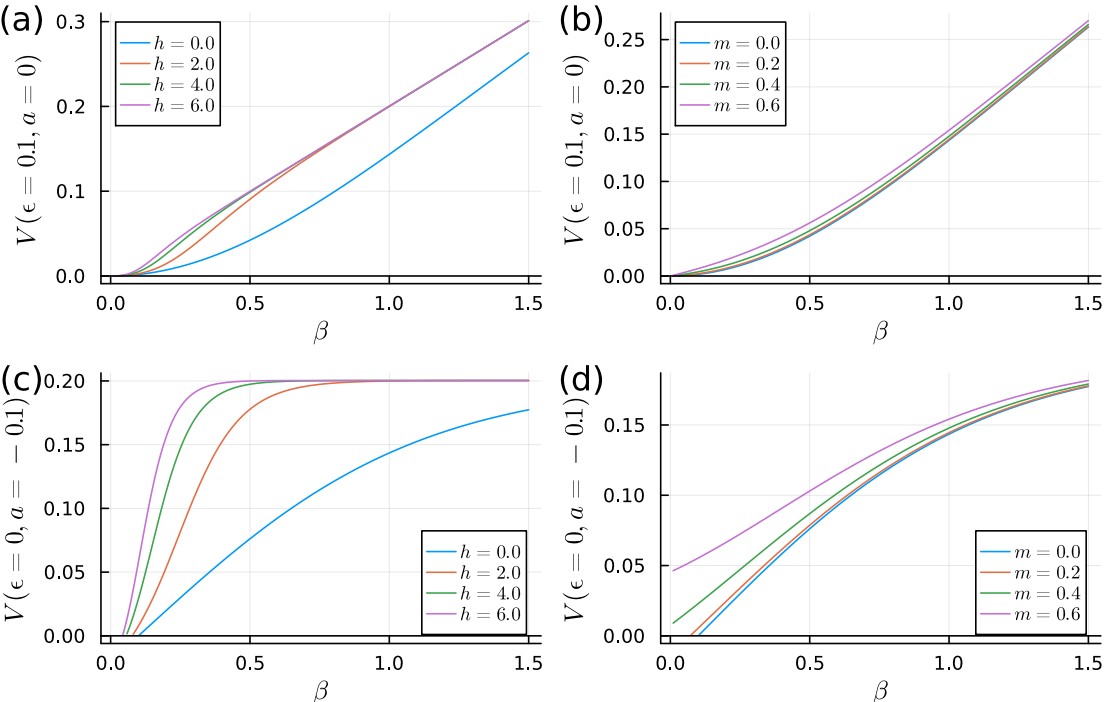

FIG. 5.   The velocity for the bond model as a function of $\beta$ for $J = 1$. Left and right columns correspond to fixed $h$ and fixed $m$, respectively. The examined values of $\varepsilon, a$ yield only motion to the right. (a-b) $\varepsilon = 0.1, a = 0$ yield $V^+(\beta = 0) = 0$ which increases monotonically with $\beta$. (c-d) $\varepsilon = 0, a = 0.1$ has a threshold $\beta = \beta^*$ beyond which $V^+ > 0$ and increases monotonically. For fixed $h$, (c), the threshold is always positive $\beta^*(h) > 0$, whereas for fixed $m$, (d), the threshold $\beta^*(m)$ becomes zero for $m > m_a(a = 0.1) \approx 0.34$, see (IV.5).

As in Section III, to analyze the conditions for vanishing velocity, we spell out the velocity to the right $V^+$, and the corresponding left-motion can be obtained by a similar analysis. The condition for nonzero velocity is unchanged, $\mu_1^{\text{bond}} < \lambda_1$, where $\lambda_1$ remains unchanged from (III.12), and

$$\mu_1^{\text{bond}} = e^{\beta J - 2(\beta\epsilon - a)}\cosh\beta h + e^{-\beta J + 2(\beta\epsilon - a)}\sqrt{1 + e^{4\beta J - 8(\beta\epsilon - a)}\sinh^2\beta h}. \qquad \text{(IV.3)}$$

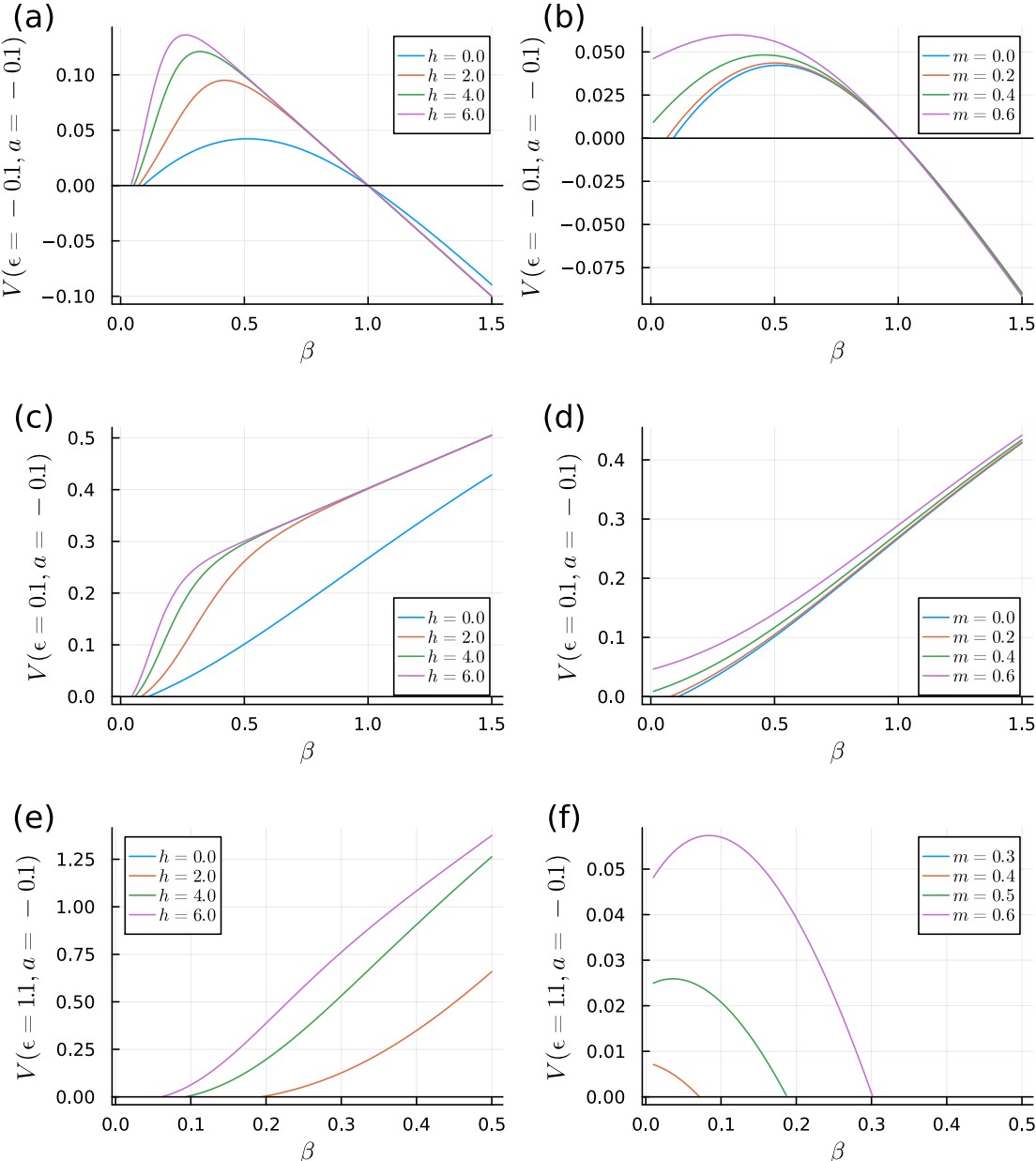

FIG. 6.   The velocity for the bond model as a function of $\beta$ for $J = 1$. Left and right columns correspond to fixed $h$ and fixed $m$, respectively. (a-b) $\varepsilon = -0.1, a = -0.1$ show competing biases resulting in non-monotonic behavior of $V^+$. For any value of $h$, (a), and for any value of $m$, (b), the velocity vanishes at $\beta_1 = a/\varepsilon$, see (IV.4), where the walker changes its direction. However, the threshold $\beta^*$ beyond which $V^+ > 0$ for both (a) and (b) depends on the values of the parameters. For $h = 0$ (equivalent to $m = 0$) it is given in (IV.4). (c-d) $\varepsilon = 0.1, a = -0.1$ yield $V^+ > 0$ that increases monotonically with $\beta$. Here, there also exists a threshold $\beta^*$ beyond which $V^+ > 0$ that depends on the parameters. (e-f) $\varepsilon = 1.1, a = -0.1$. At $h = 0$ ($m = 0$), the velocity is zero for any $\beta$. At high enough $h$, (e), the velocity becomes positive at a threshold $\beta^*(h)$ and increases monotonically with $\beta > \beta^*(h)$. For the fixed $m$ case, (f), the velocity can be strictly zero (for small $m$), have a monotonic decreasing behavior (moderate $m$), or even exhibit a non-monotonic behavior (large $m$).

It is readily seen that $\beta = \beta_1 := a/\varepsilon$ gives $\mu_1^{\text{bond}} = \lambda_1$, *i.e.*, $V^+ = 0$. On the other hand, the necessary condition (II.3) in the bond model (IV.1) reduces to $2(\beta\varepsilon - a)\langle\sigma_n\sigma_{n+1}\rangle > 0$, implying that $\beta_1$ is always a threshold inverse temperature for nonzero velocity.

For $h = 0$, $\lambda_1(h = 0) = 2\cosh(\beta J)$ and $\mu_1^{\text{bond}}(h = 0) = 2\cosh(\beta J - 2(\beta\varepsilon - a))$. As consequence, with $h = 0$, the marginal case $\lambda_1(h = 0) = \mu_1^{\text{bond}}(h = 0)$ corresponds to $\beta J = \pm[\beta J - 2(\beta\epsilon - a)]$, which yields two thresholds

$$\beta_1 = a/\varepsilon, \qquad \beta_2 = a/(\varepsilon - J), \tag{IV.4}$$

relevant only when $\beta_1 \geq 0$ or $\beta_2 \geq 0$ (and $\beta_1$ was obtained before).

The threshold inverse temperatures $\beta_1, \beta_2$ appear in Figs. 5, 6 where $J = 1$. For $\varepsilon = 0.1, a = 0$, Fig. 5(a-b), we find $\beta_1 = \beta_2 = 0$, and for $\varepsilon = 0, a = -0.1$, Fig. 5(c-d), the threshold corresponds to $\beta_2 = 0.1$, and $\beta_1$ is undefined. For $\varepsilon = -0.1, a = -0.1$, Fig. 6(a-b), the thresholds correspond to $\beta_1 = 1, \beta_2 \approx 0.09$, and for $\varepsilon = 0.1, a = -0.1$, Fig. 6(c-d), $\beta_1 = -1$ is irrelevant and $\beta_2 \approx 0.11$. For $\varepsilon = 1.1, a = -0.1$, Fig. 6(e-f), $\beta_1 \approx -0.09$ and $\beta_2 = -1$ are both irrelevant, which means that $V^+ = 0$ for all $\beta$ at $h = 0$.

Another limit that can be analyzed is $\beta = 0$, where for any fixed $h$ we get $\mu_1^{\text{bond}}(\beta = 0) = 2\cosh(2a) \geq 2 = \lambda_1(\beta = 0)$. This means that for any fixed $h$ at $\beta = 0$, the velocity is zero, as can be seen in Figs. 5(a,c), 6(a,c,e).

The case of fixed magnetization, where $h = h(\beta)$, changes this last conclusion. The analysis is done by rewriting $\lambda_1, \mu_1^{\text{bond}}$ in terms of the magnetization, using (III.15),

$$\lambda_1 = \frac{1}{\sqrt{1 - m^2}}\left(e^{\beta J}\sqrt{1 + m^2(e^{-4\beta J} - 1)} + e^{-\beta J}\right)$$
$$\mu_1^{\text{bond}} = \frac{1}{\sqrt{1 - m^2}}\left(e^{\beta J}e^{-2(\beta\epsilon - a)}\sqrt{1 + m^2(e^{-4\beta J} - 1)} + e^{-\beta J}e^{2(\beta\epsilon - a)}\sqrt{1 + m^2(e^{-8(\beta\epsilon - a)} - 1)}\right).$$

For $\beta = 0$, this reduces to

$$\lambda_1(\beta = 0) = \frac{2}{\sqrt{1 - m^2}}$$
$$\mu_1^{\text{bond}}(\beta = 0) = \frac{1}{\sqrt{1 - m^2}}\left(e^{2a} + e^{-2a}\sqrt{1 + m^2(e^{8a} - 1)}\right).$$

If $a = 0$, then at $\beta = 0$, $\mu_1^{\text{bond}} = \lambda_1$ and thus $V^+(\beta = 0) = 0$ as seen in Fig. 5(b). If $a < 0$, then the condition for $V^+ > 0$, $\mu_1^{\text{bond}} < \lambda_1$, corresponds to

$$m^2 > \frac{e^{4a}(2 - e^{2a})^2 - 1}{e^{8a} - 1} \equiv m_a^2, \tag{IV.5}$$

namely, there is some threshold magnetization $m_a$ beyond which the velocity is nonzero already at $\beta = 0$. The intuition for $m_a$ is simple. At $\beta = 0$, increasing $m$, corresponds to decreasing the number of domain walls, *i.e.*, decreasing the number of traps. $m_a$ is the threshold corresponding to this number of domain walls. See Figs. 5(d), 6(b,d,f) where $m_a(a = -0.1) \approx 0.34$. The case $a > 0$ corresponds to reversing the inequality (IV.5).

In the single bias case, Fig. 5, we observe that the velocity (to the right) increases as the temperature decreases, both for fixed Ising magnetic field $h$ and for fixed Ising magnetization $m$. This is expected as these single biases favor motion to the right, and decreasing temperature reduces the number of domain walls, hence reducing the number of bonds that favor left motion. The threshold for $V^+ > 0$ is $\beta = 0$ in the $a = 0$ case, Fig. 5(a,b), and some $\beta = \beta^* \geq 0$, for the $\varepsilon = 0$ case, Fig. 5(c,d).

For the competing biases case, Fig. 6(a,b), the velocity exhibits non-monotonic behavior. $V^+ > 0$ in the appropriate finite region of temperatures determined by (IV.4). There are a few factors at play which are responsible for the non-monotonicity. First, reducing the temperature makes neighboring spins align, which enhances both the thermal and athermal biases, $\varepsilon = -0.1$ and $a = -0.1$, respectively. At small $\beta$, the thermal bias is less relevant, and thus the athermal bias $a < 0$ enhances the motion to the right, increasing the velocity. As $\beta$ increases, the thermal bias $\varepsilon > 0$ (favoring motion to the left) becomes important, reducing the velocity to a halt at $\beta_1 = a/\varepsilon$, beyond which it is reversed, *i.e.*, the motion is to the left with $V^- < 0$.

The case of the two biases pointing in the same direction is shown in Figs. 6(c-f). Specifically, consider $a < 0, \varepsilon > 0$, both favor pushing to the right for aligned spins. Increasing $\beta$ makes the thermal bias $\varepsilon$ stronger and also lowers the number of domain walls. The naive expectation is that both effects would enhance the velocity. However, as we show below, this is not always the case. One hint for this is that while the threshold $\beta_1 < 0$, (IV.4), the threshold $\beta_2 = a/(\varepsilon - J)$ can be positive (depending on the sign of $\varepsilon - J$), suggesting there is a competition between $\varepsilon$ and $J$ that determines the onset of nonzero velocity.

Consider first the simpler case where $\varepsilon < J$ shown in Fig. 6(c,d). Here $0.1 = \varepsilon < J = 1$, so $\beta_2 > 0$, meaning there is a threshold temperature for the $h = 0$ case; see Fig. 6(c). Indeed, increasing $\beta$ increases the alignment of neighboring spins, which is aided by the thermal bias $\varepsilon > 0$ becoming more relevant, producing nonzero velocity. These effects also make the velocity $V^+$ increase monotonically with $\beta$. The thresholds $\beta^*(h)$ for the onset of

$V^+ > 0$ for $h > 0$ decrease with the magnetic field simply because it is easier to destroy domain walls when $h$ is larger. Once the magnetization saturates (which can happen only for $h > 0$), the velocity saturates to grow with the thermal bias $\varepsilon$ (see Fig. 6(a) for $h > 0$, at large enough $\beta$).

For the fixed magnetization case, still for $\varepsilon < J$, Fig. 6(d), we find a similar behavior, only that the threshold temperature $\beta^*(m)$ becomes zero at large enough magnetization $m > m_a$ (recall (IV.5)). For $\epsilon = 0$ we are in Fig. 5(d) which indeed resembles Fig. 6(d).

The case $\varepsilon > J$ corresponds to pushing hard, which might result in zero velocity. When $\varepsilon$ is large, increasing $\beta$, on the one hand, reduces the number of domain walls, but it also makes them oppose the motion more strongly. Whether the walker overcomes the opposing domain walls depends on the parameters.

In the case of $h = 0$, $\varepsilon > J$ means $\beta_2 < 0$, (IV.4), and thus, the velocity is zero for any $\beta$; see Fig. 6(e). This suggests that the rate at which domain walls are destroyed is not large enough to overcome their opposing hopping rates. Increasing $h$ further reduces the number of domain walls, and for $h > h^*$, nonzero velocity starts to appear at some $\beta^*(h) > 0$. Once there is nonzero velocity, $i.e.$, when $h > h^*$, then the velocity increases monotonically with $\beta$; see Fig. 6(e).

The fixed magnetization case, Fig. 6(f), behaves differently. As analyzed before, the value $a = -0.1$ determines the threshold magnetization $m_a \approx 0.34$, (IV.5), beyond which there is nonzero velocity at $\beta = 0$. We observe that increasing $\beta$ affects the velocity differently, depending on the magnitude of $m$. For small $m$, the velocity decreases monotonically until it vanishes, whereas for large $m$, it first increases and then decreases to zero. The case of fixed $m$, Fig. 6(f), where for large $\beta$ the velocity vanishes, is dramatically different from the case of fixed $h$, Fig. 6(e), where the velocity increases with $\beta$. The reason must be that when fixing $m < 1$, the number of domain walls decreases more slowly than when $h > 0$ is fixed. Note that in the bond model, the velocity vanishes due to a small but strong trap, in contrast to the case of the site model. In all, that is an instance of pushing harder and getting nowhere.

## V.  ADDITIONAL REMARKS

The previous detailed discussions have shown the varied and rich behavior of the asymptotic speed as a function of the (cooling) temperature, depending on parameters characterizing the bias and the environment. We add different remarks, again zooming out from the observations in the previous sections.

1. For the Ising site disorder of Section III, putting $J = 0$ makes the spins $\sigma_i$ independent. The rates (III.2) then reduce to the Derrida example in Section 4.4 in [5]. In his notation, the hopping rates are $W = e^{2(\beta\varepsilon - a)} > 1$ and 1, chosen with probability $\alpha = (1 + \tanh(\beta h))/2$ and $1 - \alpha$. There the velocity vanishes for $\alpha < \alpha^*$, for some $\alpha^* > 1/2$.

2. Consider the observation of an overdamped probe (like our walker) that accelerates. This acceleration can surely be the result of extra pushing, *e.g.*, increasing the bias in the environment. However, as shown above, acceleration can also be the result of changing correlations in an environment with a fixed bias. Hence, a proper understanding of the environment is required to resolve these two possible scenarios.

3. The scenario and results set out in the previous two sections obviously have a wider relevance. As exactly solvable models, they are useful, but they relate to a more general conclusion: the possibly rich dependence of the asymptotic speeds on the fluctuation structure in the environment. On the other hand, our probes/walkers are moving in one dimension, and trapping is obviously more common in one dimension. The fluctuation behavior may, however, be much richer in higher dimensions, as the distribution of velocities can show higher variance. Phase separation at fixed density or magnetization may cause the probe to move rapidly in one phase and slowly in the other phase. Nevertheless, the effective disorder that a probe encounters in high(er) dimensions may very well resemble the Ising-site or Ising-bond environments that have been studied explicitly in the present paper.

4. The asymptotic speed $V$ is not *thermodynamic*, in the sense that $V$ depends also on the time-symmetric parts $k(n, n+1), k(n+1, n)$ in the transition rates (and not only on the ratios $k(n, n+1)/k(n+1, n)$ that connect with work and heat, [8]). We can see

that in the presence of the matrix $S_1$, (III.10), in (III.9). That is rather normal given the case that our results are nonperturbative and include a higher-order response of the walker to multiple biases.

5. The site- and bond-disordered models exhibit qualitatively different behavior, as can for instance be seen by the fixed magnetization scenario shown in Fig. 2 and Fig. 5(b,d). In the bond-disorder model, Fig. 5(b,d), the speed is always increasing for lower temperatures and for higher $m$. That is certainly not the case for the site-disorder; see Fig. 2. The difference is, of course, that for the bond-disorder, the walker is possibly slowed down at domain walls, not in the bulk of domains. At lower temperatures, the domain walls decrease. For the Ising-site disorder, the bulk of the domains matters.

6. The fact that fluctuation behavior, correlations, and phase separation play a role in the behavior of the speed of the walker in a random environment is not surprising. Homogenization (like in (III.4) or mean-field like treatments) would ignore the correlations and miss the essential physics.

As a final remark we dare to suggest an analogy with the famous *dark-energy problem*, [9–16]. That the expansion of the Universe is accelerating has obtained evidence around 1998-99 from the observations of high-redshift Type Ia supernovæ, [17]. Stars and galaxies keep their shape but the distance between them is growing at an increasing rate. In fact, one could have expected a *deceleration* under the influence of gravity, which at first sight competes with the expansion (anti-gravity effect). In addition, the so-called "coincidence or fine-tuning problem" arises, of why given the enormous age of the universe, only recently (at the time of structure formation) gravity and anti-gravity effects are of comparable sizes.

The possible (only conceptual) relation of our work with that so-called "dark energy" problem is obtained by thinking of the distance between galaxies as the distance traveled by our walker in a disordered environment. As the environment cools, the competition between the biases (denoted by $\varepsilon$ (expansion) and $a$ (attraction) in the models above) changes, and the rate of expansion may change, *e.g.*, yielding acceleration. That point is missed in a fully homogenized analysis, as *e.g.* in the Friedmann-Lemaître-Robertson-Walker equations. Including fluctuations requires a nonperturbative analysis away from a mean-field analysis. In that sense, our suggestion is in

line with the backreaction-program [18–21], except that, for the discussion of cosmic expansion, our modeling is technically simplified.

## VI.   CONCLUSIONS

A random environment is obviously more than its density or magnetization. Fluctuations and correlations matter for the speed of the walker, producing a very rich variety of possibilities. We have illustrated that point in considerable detail via exact solutions when the environment is made from the one-dimensional Ising model. We have considered site- and bond-versions, and with thermal and athermal biases. The dependence of velocity on temperature at fixed magnetization shows a number of regimes, from which we observe deceleration or acceleration as correlation lengths change. As the environment cools and depending on the competition between the biases, the walker will change its (asymptotic) speed, from zero to a positive value and will show various non-monotone behaviors.
The results explicitly state the case for a rich and varied velocity characteristic as long as the environment is not homogenized.

**Acknowledgment**: This work was started while the authors shared an office at the Erwin Schrödinger Institute in Vienna. We acknowledge the stimulating atmosphere there during the Thematic Programme "Large Deviations, Extremes and Anomalous Transport in Non-equilibrium Systems" from September 19 to October 14, 2022.

---

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
