# Peer review of "Acceleration from a clustering environment"

_SciPost Physics_

## Round 1 · Referee Report · Anonymous (Referee 1) · 2024-7-31

Strengths

1) The impact of correlations in the random environment of a random walker is studied on a simple model in which the environment is generated from an Ising model.

2) A parallel is made with the dark-energy problem: the current understanding of the expansion of the universe is based on a mean-field analysis and the authors raise the point that the study of correlations might play a role.

3) The manuscript is well written and pleasant to read.

Weaknesses

1) The analysis is limited to the 1D case, which is known to be the geometry in which disorder has the most significant effects.

2) Most results come from an application of a formula derived by Derrida (J. Stat. Phys. 1983) to the specific model introduced in the manuscript.

Report

In this manuscript, the authors study the asymptotic speed of a random walker in a random environment. They consider a simple 1D model in which the environment is generated from a realisation of the Ising model and thus presents correlations. They discuss the impact of these correlations on the speed of the walker by fixing the magnetisation but varying the correlation length. They discover a variety of nontrivial behaviours, and in particular show that a cooling of the environment (with the magnetisation still fixed) can yield an acceleration of the walker. A parallel is drawn with the dark-energy problem and the acceleration of the expansion of the universe.

The paper is interesting, clear and well written, although some points could still be clarified (see below). The model is simple but has the advantage to yield exact results that are thoroughly examined. From a purely technical point of view, the results are obtained from a known formula of Derrida (J. Stat. Phys. 1983) and standard calculations on the Ising model. The interest of the manuscript relies on the analysis of these exact formulas and their physical implication. I am not qualified to judge if the parallel with the acceleration of the universe is relevant, but from an outsider's point of view, it seems reasonable to question the importance of correlations in this context. Finally, this paper raises many questions that call for follow-up work.

Requested changes

1) When the model is introduced in Section III.A, it would be useful to introduce explicitly the inverse temperature $\beta$ below Eq. (III.1) as the one of the Ising model, which thus controls the correlation length (recalled on page 9). At the moment it is not so clear in Section III.A that $\beta$ is a parameter of the Ising model.

2) Related point: in Eqs. (III.2) and (III.3) why does $\beta$ appear in the jump rates of the random walker? I would naively think there should be two different temperatures: the one controlling the environment (Ising) and the one controlling the walker. Why are they chosen to be related (via the parameters $a$ and $\epsilon$)?

3) The implication of the results for a "cooling environment" is discussed several times. However, all the study is conducted for a random walker in a given environment, so that these discussions assume that the temperature is changed adiabatically. This should be written explicitly (at the moment it is only briefly mentioned in the introduction).

4) In the second paragraph of the introduction, it is mentioned that a rich phenomenology is observed in a "cooling environment". At this stage it is not clear why the authors only consider the case of cooling. The motivation is implicit in the next paragraph (the universe is cooling down), but the case of increasing temperature could also be interesting from a stat mech point of view. Could the authors comment on that?

Recommendation

Ask for minor revision

  • validity: top
  • significance: high
  • originality: high
  • clarity: high
  • formatting: excellent
  • grammar: excellent

Author:  Roi Holtzman  on 2024-08-22  [id 4711]

(in reply to Report 1 on 2024-07-31)

We thank the referee for the detailed report.

Below, we address the requested changes.

1) Thanks to this comment, we now mention that the Ising model is at that inverse temperature β.

2) The referee is absolutely correct that, generally, one could choose different temperatures for the hoppings of the walker and the environment. Our parameterization allows for the same generality while, at the same time, providing some natural choice that is able to tune the environment and the walker in a corresponding manner. Indeed, taking a different temperature for the walker amounts to setting ϵ=0 and modifying $a$ as a temperature. In the paper, we fix $a$ in all cases and discuss the dependence of the walker on the environment's temperature, as this is the main effect we are interested in. We have pointed this out clearly when defining the hopping rates.

3) We thank the referee for this comment. We have delayed the mentioning of a "cooling environment" to Section III.A. and specified explicitly what we mean. Indeed, we interpret the 'cooling' in the quasistatic sense.

4) There is indeed no reason to restrict that comment to 'cooling' albeit more relevant for cosmological considerations. We have added a comment/remark.

---

## Round 1 · Referee Report · Anonymous (Referee 2) · 2024-8-7

Strengths

  1. Efficient analysis.
  2. Well-organized and well-written MS.
  3. Speculative, but intriguing thoughts about similar mechanisms relevant in cosmology.

Weaknesses

  1. Cooling is mentioned in the Abstract and Introduction, but similarly not developed in the technical part of the MS.
  2. The analysis is restricted to the one-dimensional toy models. The implications for cosmology developing in the three-dimensional space are uncertain even if one ignores the lack of other crucial ingredients (gravity etc.).

Report

The authors study the effects of correlations in a random environment on a random walker. There are hundreds of studies of similar problems, yet the MS is surprisingly refreshing, and the authors find several new phenomena and unexpected features. The intriguing observed phenomenon emphasized by the authors is acceleration, and they provide bold speculations on the possible relevance of cosmic acceleration. One could imagine different reactions from the readers. I bet this speculative connection energized the authors. The outcome is a well-written and inspiring MS. I may be biased since I share the authors' view that some recent insights from non-equilibrium statistical physics, such as astounding (in some settings) difference between annealed and quenched disorder, may have cosmological implications.

A minor remark concerns cooling, mentioned in the Abstract and Introduction, but not in the main body of the paper. Also, whenever readers with a background in statistical physics hear about cooling and cosmology, they inevitably remember the Kibble-Zurek mechanism (KZM). Such readers may contemplate the similarities and dissimilarities of the KZM and the effects of the cooling that the authors have in mind.

My main concern is that the rich phenomenology appearing in one dimension would be absent in three dimensions. The dominant role of disorder in low-dimensional systems is well-known, and it happens for many random walk problems. The simplest example is the fate of a stationary target particle in the infinite sea of non-interacting random walkers (or interacting like the SEP). The target particle survives till time $t$ if it has not been hit by a random walker. In one dimension, there is a drastic difference between the surviving probability in the annealed and quenched settings; in three spatial dimensions, there is no difference (as far as the leading asymptotic is concerned). The evolving systems with never-ending memory of the details of the (statistically uniform) initial conditions exist, but they are rare. I would not be surprised if the authors contemplated these issues, and I recommend sharing their thoughts even if they are impressionistic. The penultimate section *Additional Remarks* is a perfect place for such comments.

Overall, I recommend asking the authors to think about the above remarks and perhaps make minor revisions.

Recommendation

Ask for minor revision

  • validity: high
  • significance: good
  • originality: high
  • clarity: high
  • formatting: excellent
  • grammar: excellent

Author:  Roi Holtzman  on 2024-08-22  [id 4712]

(in reply to Report 2 on 2024-08-07)

We thank the referee for the detailed report.

In our revised version we have included comments about the raised points.

We have introduced the scenario of a cooling environment explicitly in Section III.A stating that this means a quasistatic decrease of the inverse temperature β. Moreover, thanks to the referee's comment, we have mentioned the Kibble-Zurek mechanism in remark 3 in Section V.

As the referee pointed out, dimensionality plays an important role in disordered systems, particularly in the differences between quenched and annealed averages. Our analysis serves as an illustration of the rich effects correlations in an environment may have on observables of a probe. It would be interesting to see how dimensionality affects the results in future work. We have stressed this in remark 3 in Section V. Regarding the implications to cosmology, our modeling is far from the physical system of the universe, and a three-dimensional version would not make it substantially more realistic.

---

## Editorial Decision

resubmitted